# Pharmacological Activation of Potassium Channel Kv11.1 with NS1643 Attenuates Triple Negative Breast Cancer Cell Migration by Promoting the Dephosphorylation of Caveolin-1

**DOI:** 10.3390/cells11152461

**Published:** 2022-08-08

**Authors:** Ying Jiang, Vitalyi Senyuk, Ke Ma, Hui Chen, Xiang Qin, Shun Li, Yiyao Liu, Saverio Gentile, Richard D. Minshall

**Affiliations:** 1Center for Informational Biology, University of Electronic Science and Technology of China, Chengdu 610054, China; 2Department of Pharmacology, University of Illinois at Chicago, Chicago, IL 60612, USA; 3Department of Medicine, University of Illinois at Chicago, Chicago, IL 60612, USA; 4Research Resources Center, University of Illinois at Chicago, Chicago, IL 60612, USA; 5UI Cancer Center, University of Illinois at Chicago, Chicago, IL 60612, USA; 6Department of Anesthesiology, University of Illinois at Chicago, Chicago, IL 60612, USA

**Keywords:** calpain, PTP1B, β-catenin, R-cadherin, adherens junction, focal adhesion complex, triple negative breast cancer metastasis

## Abstract

The prevention of metastasis is a central goal of cancer therapy. Caveolin-1 (Cav-1) is a structural membrane and scaffolding protein shown to be a key regulator of late-stage breast cancer metastasis. However, therapeutic strategies targeting Cav-1 are still lacking. Here, we demonstrate that the pharmacological activation of potassium channel Kv11.1, which is uniquely expressed in MDA-MB-231 triple negative breast cancer cells (TNBCs) but not in normal MCF-10A cells, induces the dephosphorylation of Cav-1 Tyr-14 by promoting the Ca^2+^-dependent stimulation of protein tyrosine phosphatase 1B (PTP1B). Consequently, the dephosphorylation of Cav-1 resulted in its disassociation from β-catenin, which enabled the accumulation of β-catenin at cell borders, where it facilitated the formation of cell–cell adhesion complexes via interactions with R-cadherin and desmosomal proteins. Kv11.1 activation-dependent Cav-1 dephosphorylation induced with NS1643 also reduced cell migration and invasion, consistent with its ability to regulate focal adhesion dynamics. Thus, this study sheds light on a novel pharmacological mechanism of promoting Cav-1 dephosphorylation, which may prove to be effective at reducing metastasis and promoting contact inhibition.

## 1. Introduction

Ion channels have recently been shown to play important roles in cancer [1], including metastasis [2,3,4,5]. Kv11.1 is a surface membrane potassium ion channel that was shown to be expressed in a variety of cancers of different histogeneses [6,7,8]. Kv11.1 channels appear to affect numerous mechanisms critical to cancer survival, ranging from cell proliferation to metastasis [9,10,11]. We previously demonstrated that Kv11.1 activator molecules (e.g., NS1643, PD115087) inhibit metastasis by arresting the motility of colon and breast cancer cells [12,13]. In those static cancer cells, we detected accumulations of both E-cadherin and β-catenin at the plasma membrane, as well as the inhibition of β-catenin-dependent transcription, indicating a clear pattern of cell–cell contact inhibition and the reversal of EMT (epithelial to mesenchymal transition), which contributes to an inhibition of cancer cell motility. However, whether the same effects extend to highly metastatic breast cancers that lack E-cadherin expression is unclear. Therefore, studying the role of Kv11.1 activation on the migratory ability of highly metastatic and mesenchymal-like triple negative breast cancer cells is of great interest for its clinic application.

Caveolin-1 (Cav-1), an integral membrane protein required for caveolae formation that also functions as a scaffolding protein and a key regulator of cell signaling molecules [14], was shown to promote tumor cell migration, invasion, and metastasis in breast, colon, lung, bladder, prostate cancer, melanoma, and others [15,16,17,18,19,20]. The underlying mechanism is related to its phosphorylation on the tyrosine 14 residue, which was shown to regulate its binding to signaling molecules [21,22], including those that regulate tumor cell migration [23]. pY14-Cav-1 closely associates with the cytoskeleton system. Joshi et al. [24] demonstrated that Cav-1 localized to the protrusive domains of tumor cells, where it was phosphorylated by Src and associated with Rho/ROCK signaling. Phosphorylated Cav-1 reciprocally enhances RhoA activity and facilitates focal adhesion turnover, leading to enhanced tumor cell migration. Cytoplasmic Cav-1 was shown to be associated with contractile actin filaments [25]. These authors showed that cytoplasmic Cav-1-positive vesicles move along actin filaments, but once Cav-1 is dephosphorylated, the motility of these vesicles is decreased, due to reduced RhoA-myosin Ⅱ activity and increased Rac1-PAK1-Cofillin activation, resulting in the disorganization of contractile stress fibers and compromised directional movement. Moreover, Cav-1 was found to interact with E-cadherin and β-catenin, and was able to inhibit β-catenin-Tcf/Lef-dependent transcription, leading to reduced cyclooxygenase-2 and survivin expression, and enhanced metastasis [26,27,28]. The expression levels of Cav-1 and β-catenin were found to be highly correlated in breast cancer cells after exposure to chemotherapy, both in vitro and in vivo [29]. Cadherins and catenins form complexes that make up adherens junctions that stabilize cell–cell contacts. How Cav-1 expression affects junctional integrity and promotes contact inhibition in association with cadherins or catenins remains poorly characterized in cancer cells.

Here, we demonstrate a novel signaling mechanism for inhibiting cell migration and promoting the cell–cell adhesion of MDA-MB-231 triple negative breast cancer cells that lack expression of E-cadherin. We show that phosphorylated Cav-1 plays an essential role in negatively regulating cell–cell contact by associating with/sequestering β-catenin and promoting cell migration. β-catenin dissociation from non-phospho-Cav-1 exhibited greater interaction with R-cadherin and desmosomal proteins, as shown by the untargeted proteomics analysis of proteins pulled down with β-catenin in NS1643 treated cells. Thus, in addition to adherens junctions, these data also indicate that desmosomes actively participate in the accumulation of β-catenin at cell–cell junctions to promote contact inhibition. Therefore, we discovered that the dephosphorylation of Cav-1 can be achieved using pharmacological stimulation of the potassium channel Kv11.1, which is uniquely and highly expressed in MDA-MB-231 triple negative breast cancer cells. Kv11.1 activating compounds increase intracellular Ca^2+^, thereby promoting calpain cleavage, the activation of PTP1B, and the subsequent dephosphorylation of Cav-1 Y14. To our knowledge, this is the first observation of potassium channel activation-dependent Cav1 dephosphorylation in highly metastatic breast cancer cells, which importantly, may be an effective therapeutic strategy for inhibiting migration and promoting the contact inhibition of metastatic cancers.

## 2. Materials and Methods

### 2.1. Cell Culture

MDA-MB-231 and MCF-7 cell lines purchased from ATCC (American Type Culture Collection, Manassas, VA, USA) were cultured in DMEM growth media (Corning, Corning, NY, USA) supplemented with 10% FBS, 10 mM HEPES, 50 U/mL penicillin, and 50 μg/mL streptomycin.

### 2.2. Reagents

All reagents were obtained from Sigma-Aldrich (St. Louis, MO, USA) unless stated otherwise. NS1643 was from Alamo Laboratories Inc. (San Antonio, TX, USA). Calpeptin was purchased from Tocris Bioscience (Minneapolis, MN, USA). PTP1B inhibitor was from Cayman Chemical (Ann Arbor, MI, USA). Phospho-Y14, total Caveolin-1, β-catenin, and caspase-3 antibodies were from BD Biosciences (San Jose, CA, USA). Phospho-Y416, Y527, and total Src were from Cell Signaling Technology (Danvers, MA, USA). PTP1B, desmoplakin, desmoglein, and plakophilin monoclonal Abs, normal mouse IgG, and protein A/G agarose beads were obtained from Santa Cruz Biotechnology, Inc. (Santa Cruz, CA, USA). DAPI and all fluorescently labeled secondary antibodies were purchased from Molecular Probes (ThermoFisher Scientific, Waltham, MA, USA). Active β-catenin monoclonal antibody was purchased from MilliporeSigma (Burlington, MA, USA). HRP-conjugated goat-anti-mouse and goat-anti-rabbit secondary antibodies were from KPL (Gaithersburg, MD, USA). Lipofectamine 2000, ECL Super Signal kit, and Crosslink magnetic IP/Co-IP kit were from ThermoFisher Scientific (Waltham, MA, USA).

### 2.3. Transfection and Infection

Caveolin-1 adenovirus was generated in the Viral Vector Core Laboratory at UCSD as described [30]. Full-length *Homo sapiens* caveolin-1 was used as a template to generate C-terminal CFP-tagged caveolin-1 (Cav-1-CFP), as well as the phospho-defective and mimicking mutants Y14F and Y14D [31]. β-catenin-Venus plasmid was a gift from Dr. Andrei Karginov (UIC). pRVH retroviral vector encoding Cav-1 shRNA was provided by Dr. William A. Muller (Northwestern University). MDA-MB-231 cells were infected with Cav-1 shRNA retrovirus to generate stable Cav-1 depleted cell lines. After overnight culture, the virus-containing medium was replaced with fresh complete medium, and cells were used for experiments, 48–72 h after infection. MCF-7 cells or stable Cav-1 depleted MDA-MB-231 cells transfected with Cav-1 WT, Y14F, and Y14D mutants, with or without CFP-tag, using Lipofectamine 2000 for 48–72 h were visualized using confocal microscopy, lysed for Western blot, or fixed for immunostaining.

### 2.4. Western Blot Analysis and Immunoprecipitation 

For Western blotting, cells after treatment were lysed on ice and sonicated 3 times for 10 s each in RIPA buffer containing protease inhibitor cocktail. All insoluble material was removed via centrifugation (16,200× *g* for 15 min). The protein concentration of the supernatant was determined using a BCA Protein Assay kit. After being prepared in sample buffer (50 mM Tris pH 6.8, 2% SDS, 6% glycerol, 0.01% bromophenol blue) supplemented with 10 mM DTT and boiled for 5 min, samples were then subjected to SDS-PAGE. Separated proteins were transferred from the gel to a nitrocellulose membrane via wet electroblotting. Nitrocellulose membranes were then incubated with primary and secondary antibodies and processed with an ECL Super Signal kit, and then relative band intensities from the scanned images (densitometry) were determined using ImageJ.

For immunoprecipitation, cells were suspended with 2% octyl-D-glucoside (ODG) in Tris Buffer (20 mM Tris-HCl, 150 mM NaCl, and 1 mM EDTA; pH 7.4) containing protease inhibitor cocktail and disrupted via sonication 3 times for 10 s each. After removing insoluble material, cell lysates were incubated with polyclonal anti-Cav-1 antibody overnight at 4 °C, followed by incubation with protein A/G agarose beads for 1 h at RT (room temperature). Beads were collected via centrifugation and washed with 1% ODG Tris Buffer 10 times. Proteins were then eluted by adding 50 μL of sample buffer and boiling for 5 min prior to Western blotting.

### 2.5. Immunostaining and Microscopy

For immunostaining, MDA-MB-231 or Cav-1 depleted cells treated with DMSO or 50 μM NS1643 for 72 h were fixed with 4% paraformaldehyde (PFA) in PBS for 20 min at RT. Fixed samples were blocked with blocking buffer (2% BSA and 0.05% Tx-100 in PBS) for 90 min at RT and then incubated with β-catenin primary antibody overnight at 4 °C, followed by secondary antibody incubation for 1 h at RT. The nuclei were stained with DAPI in PBS containing 2% BSA. Slides were washed with PBS containing 0.01% Tween-20 at least 6 times. After mounting with Prolong diamond antifade reagent, slides were visualized using Zeiss LSM880 confocal microscope (Carl Zeiss MicroImaging, Inc., White Plains, NY, USA).

For live cell imaging, Cav-1 depleted MDA-MB-231 cells were transfected with β-catenin-Venus alone or in a combination of CFP-tagged Cav-1 WT, Y14D, or Y14F cDNAs for 48 h, and imaged with a Zeiss LSM710 META confocal equipped with a PeCon heated stage and a CO_2_ controller. Image sequences were acquired using 458 and 514 nm excitation laser lines and pinhole set to achieve 1 Airy unit. Image analysis was performed using ImageJ, ImagePro plus 6.0, and Imaris Image Analysis Workstation (Oxford Instruments plc, Abingdon, UK).

### 2.6. Scratch-Wound Assay

MDA-MB-231 cells with and without 48 h exposure to Cav-1 shRNA were seeded on 35 mm dishes at 0.8 × 10^6^ cells per well for 24 h. When cells were at a suitable confluence, a 200 μL pipette tip was used to scratch a wound through the center of the monolayer. Images were acquired with a Nikon Eclipse E600 Fluorescence Microscope (Nikon Inc., Minato, Japan) at 0, 14 and 24 h after scratch wounding. The wound area was measured using ImageJ. Wound closure rates (R_wc_) were calculated using the equation R_wc_ = (A_0_ − A_t_)/A_0_, where A_0_ represents the wound area at time 0 and A_t_ represents the wound area at corresponding time points. 

### 2.7. Single Cell Tracking

MDA-MB-231 or MCF-7 cells transfected with CFP-tagged Cav-1 WT, Y14D, Y14F mutants were plated into glass-bottom dishes at a density of 5000 cells per well. After 24 h, time-lapse images were acquired using an Olympus VivaView FL Incubator Microscope every 6 min for 4 h. Image sequences were analyzed using ImageJ software with a manual tracking plugin.

### 2.8. Spheroid Generation and Growth

A 50 μL volume of hot 1.5% agarose was added to each well of a 96-well round bottom plate. After solidification, 1000 MDA-MB-231 or Cav-1 depleted cells in 20 μL culture media containing 10% Matrigel were plated into each well. At 48 h after plating, 100 μL medium per well was removed and discarded, and 100 μL fresh medium containing 10% Matrigel was added. After culturing for one week with medium changes every 2–3 days, spheroids were incubated with DMSO or 50 μM NS1643 and visualized using a Nikon Eclipse E600 fluorescent microscope at time 0 and 24 h. Images were processed using ImageJ. Spheroid volume was calculated using the equation V = (w^2^ l)/2 (width, w and length, l).

### 2.9. TMT Labeling and Mass Spectrometry

A crosslinking magnetic IP/Co-IP kit was used to immunoprecipitate β-catenin binding proteins. First, the β-catenin antibody was covalently crosslinked to Protein A/G magnetic beads according to the manufacturer’s instructions. Briefly, 4 μg/100 μL of antibody was added to pre-washed magnetic beads and incubated for 30 min on a rotating platform. Beads were then collected using a magnetic stand and washed 3 times with modified coupling buffer. A total of 20 μM DSS was then incubated with the beads for 30 min to promote covalent crosslinking. Beads were collected and 100 μL elution buffer was added to remove non-crosslinked antibody. Beads were then washed 4 times with the modified coupling buffer. 

MDA-MB-231 cells treated with DMSO or 50 μM NS1643 for 72 h were lysed with IP lysis/wash buffer and centrifuged at 16,200× *g* for 15 min to pellet cell debris. The supernatant was incubated with magnetic beads crosslinked with β-catenin antibody overnight at 4 °C. Beads were collected and washed 6 times with IP lysis/wash buffer, and then eluted with elution buffer. The immunoprecipitated proteins were subjected to SDS-PAGE and the gel was stained with Coomassie blue to assess the relative amount of specifically bound and pulled-down proteins.

The samples were then labeled with the TMT6plex Isobaric Label Reagent Set, following manufacturer’s instructions. The scheme for the TMT labeling of 6 samples is shown in Appendix A. Briefly, 10 μg sample was re-suspended in 50 μL 50 mM triethylammonium bicarbonate (TEAB) buffer, and approximately 80 μg TMT reagents (Channel 1 to Channel 6) in 20.5 μL anhydrous acetonitrile was added into each sample. After 2 h incubation at RT, the reaction was quenched with 4 μL of hydroxylamine for 15 min. The combined equal amount of each batch was dried and desalted with a primed HLB 96-well plate for high-pH reverse phase (HPRP) fractionation. The labeled peptides were separated into 70 fractions using a XBridge BEH C18 Column (130 Å, 3.5 μm, 4.6 mm × 250 mm; Waters). Every 10 fractions were combined, and 10 concatenated fractions were dried and resuspended in 12 μL of 5% acetonitrile and 2% formic acid buffer. A total of 6 μL of the concatenated HPRP fractions were analyzed using a Q Exactive HF mass spectrometer coupled with an UltiMate 3000 RSLC nanosystem with a Nanospray Frex Ion Source (ThermoFisher Scientific, Waltham, MA, USA). Data were analyzed using Ingenuity Pathway Analysis (QIAGEN, Venlo, The Netherlands).

### 2.10. Statistical Analysis

Data are presented as Mean ± SD of at least three independent experiments. Results were analyzed with GraphPad Prism9 using one-way ANOVA for multiple group comparisons, two-way ANOVA for pairwise multiple comparisons, and Student’s *t* test for two group comparisons; a threshold of *p* < 0.05 was considered statistically significant. Survival probability was analyzed using the Kaplan–Meier log-rank test.

## 3. Results

### 3.1. Cav1 Mediates the Kv11.1 Activation-Dependent Inhibition of Cancer Cell Migration

We first investigated whether Cav1 expression contributes to the inhibitory effects of Kv11.1 activation on cancer cell migration. A wound-healing assay was performed before and after treatment with NS1643 for 24 h (Figure 1A,B) in MDA-MB-231 breast cancer cells treated with control or Cav1-specific shRNA. The results indicate that NS1643 significantly decreases the migratory ability of MDA-MB-231 in naïve but not Cav-1 depleted cells.

As the rate of wound healing is regulated by both proliferation and cell motility, we next separately assessed proliferation and cell motility by first generating 3D cultures of MDA-MB-231 spheroids. One week after spheroid generation, NS1643 was applied for 24 h. We observed that NS1643 treatment reduced the proliferation of cells expressing Cav-1; however, when Cav-1 expression was downregulated, the inhibitory effect was no longer detected, as shown in Figure 1C,D.

To study cancer cell motility in a quantitative manner at a single cell level, we took advantage of a single cell tracking method. The tracking analysis of MDA-MB-231 cells exposed to NS1643 for 6 h demonstrated that NS1643 significantly inhibited both the velocity and distance traveled of moving cells (Figure 1E,F). Quantitative analysis showed that cells exposed to NS1643 exhibited a reduced velocity (a median value of 0.29 µm/min in control cells versus 0.20 µm/min in NS1643 treated cells) and a shorter distance traveled (a median value of 110 µm in control cells versus 70 µm in NS1643 treated cells). No reduction in either velocity or distance was observed in Cav-1 depleted cells treated with NS1643. These data suggest Cav1 plays a primary role in mediating the inhibitory effect of Kv11.1 activation on cancer cell migration.

**Figure 1 cells-11-02461-f001:**
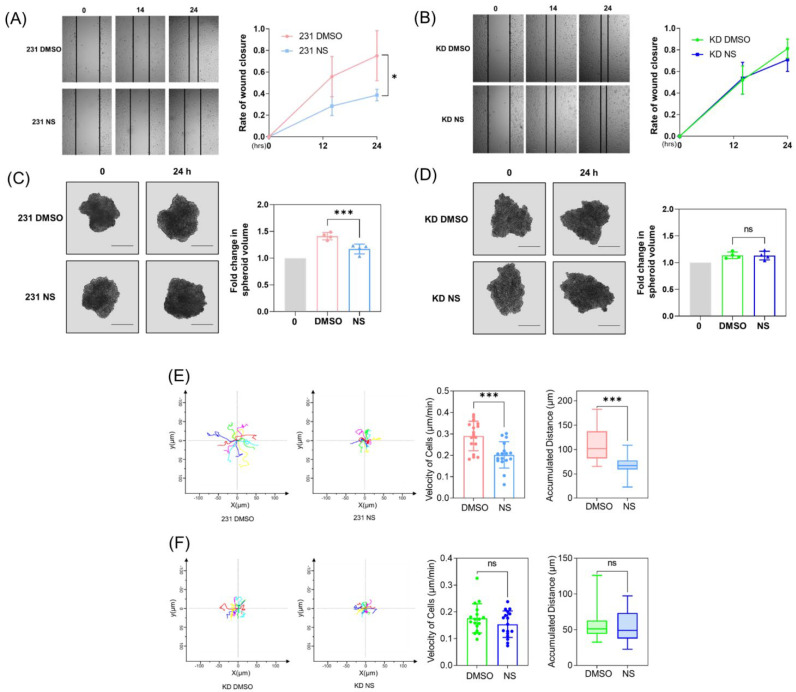
Kv11.1 activator NS1643 inhibits cell migration and growth in a manner dependent on Cav-1 expression. (**A**) MDA-MB-231 cells were scratch wounded and treated with DMSO or 50 μM NS1643 for 24 h. The rate of wound closure was significantly reduced by NS1643, n = 3. (**B**) Cav-1 depleted MDA-MB-231 cells (KD) were scratch wounded and treated with DMSO or 50 μM NS1643 for 24 h. The rate of wound closure was not affected by NS1643 in absence of Cav-1, n = 3. (**C**) Spheroids, generated by seeding 1000 MDA-MB-231 cells and cultured for one week, were treated with DMSO or 50 μM NS1643 for 24 h. Spheroid volume was reduced using NS1643. Scale bar = 500 μm, n = 4. (**D**) Spheroid volume was not affected by NS1643 in Cav-1 downregulated spheroids. Scale bar = 500 μm, n = 4. ns = not significant, * *p* < 0.05, *** *p* < 0.001 via ANOVA. (**E**) MDA-MB-231 cells seeded on glass-bottom dishes were treated with DMSO or 50 μM NS1643, and images were acquired every 6 min to track the movement of individual cells for 6 h. Image sequences were processed using ImageJ. NS1643 decreased cell movement velocity and distance migrated, n ≥ 15. (**F**) The motility of individual Cav-1 depleted MDA-MB-231 cells was traced, and no effect was detected in NS1643 treated groups, n ≥ 15. All data are mean ± SD. ns = not significant, *** *p* < 0.001 via *t* test.

### 3.2. Kv11.1 Activation Decreases Cav1 Phosphorylation Independent of Src

Because the phosphorylation of Cav1 on tyrosine-14 (pY14-Cav1) is closely associated with cell migration [32], we monitored the effect of activation of Kv11.1 on the phosphorylation status of Y14-Cav1. Western blot analysis revealed that treatment of MDA-MB-231 with NS1643 resulted in a significantly reduced phosphorylation of Cav1 Y14, as shown in Figure 2A. A time course experiment showed that the effect of NS1643 on pY14-Cav1 was rapid and persistent (Figure 2B). Furthermore, the NS1643-dependent dephosphorylation of pY14-Cav1 also occurred when Cav1 was heterologously overexpressed in MCF7 breast cancer cells (Figure 2C).

It has been well established that Y14-Cav1 is a primary Src substrate in many cell types [33]. Additionally, Src kinase activity is regulated by the phosphorylation status of Y416 (pY416-Src; active) and Y527 (pY527-Src; inactive) [34], such that the ratio of Src Y416 and Y527 phosphorylation can serve as a surrogate indicator of Src activity. Interestingly, we found that NS1643 treatment reduced the phosphorylation levels of both Y416 and Y527 residues, especially after long-term treatment (over 48 h; Appendix A). However, the reduction in pY527 was greater than that of pY416, suggesting if anything, that Src activity may have increased in cells treated with NS1643 (Appendix A), whereas the phosphorylation of Cav-1 continuously decreased in the presence of NS1643. These data indicate that the inhibitory role of NS1643 on Cav-1 phosphorylation is independent of Src activity.

**Figure 2 cells-11-02461-f002:**
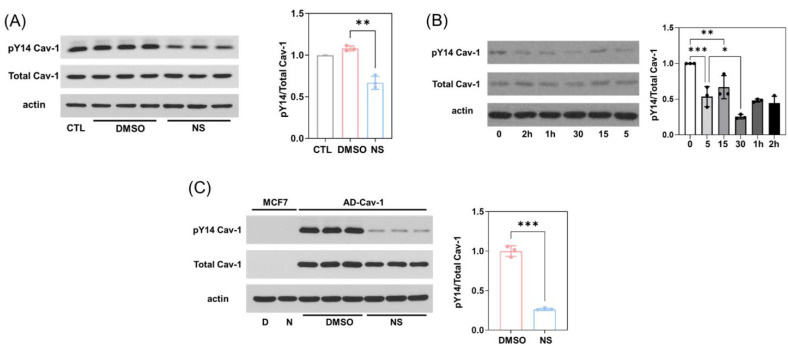
Kv11.1 activator NS1643 promotes dephosphorylation of Cav-1 but not through inhibition of Src activity. (**A**) MDA-MB-231 cells treated with DMSO or 50 μM NS1643 for 24 h were lysed and subjected to Western blotting. The treatment of NS1643 significantly reduced the phosphorylation level of Cav-1 without changing the total Cav-1 protein level. (**B**) MDA-MB-231 cells treated with 50 μM NS1643 for 0, 5, 15, or 30 min, or 1 or 2 h were collected, lysed, and subjected to Western blotting. The reduction in phosphorylation of Cav-1 induced by NS1643 was observed at 5 min and intensified over time. * *p* < 0.05, ** *p* < 0.005, *** *p* < 0.001 via ANOVA. (**C**) MCF-7 cells infected with adenovirus encoding Cav-1 were treated with DMSO or 50 μM NS1643 for 24 h, and then lysed and subjected to Western blotting. NS1643 significantly reduced the phosphorylation level of exogenously expressed Cav-1. n = 3; *** *p* < 0.001 via *t* test. All data are mean ± SD.

### 3.3. Kv11.1 Activation of PTP1B Leads to the Dephosphorylation of Cav1 Y14

We previously demonstrated that the stimulation of Kv11.1 activity produces significant Ca^2+^ entry in breast cancer cells [35]. To identify the molecular mechanism linking pharmacologic Kv11.1 activation with the dephosphorylation of Cav1, we tested the hypothesis that NS1643-induced dephosphorylation of Cav1 is related to a Ca^2+^-activated tyrosine phosphatase. 

We first studied the phosphorylation status of Cav1 in cells treated with NS1643, alone or in combination with the Ca^2+^ chelator EGTA, which blocks Ca^2+^ entry. We found that presence of EGTA strongly inhibited dephosphorylation events induced by NS1643 on both pY14-Cav1 and pY416 Src, as shown in Figure 3A. 

In contrast, the use of Ca^2+^ ionophore A23187, which increases intracellular calcium levels, significantly induced the dephosphorylation of Cav1 which could be partly restored by the calpain inhibitor calpeptin (Figure 3B). As calpain is a calcium-dependent cysteine protease expressed ubiquitously in mammals and can mediate the cleavage and activation of Protein Phosphotyrosine Phosphatase 1B (PTP1B), we next explored the involvement of PTP1B as a Ca^2+^-dependent effector. We first treated cells with A23187 in combination with PTP1B inhibitor (PI) and assessed Cav-1 phosphorylation and PTP1B cleavage. PI was capable of partially rescuing PTP1B cleavage and Cav-1 dephosphorylation induced by A23187 (Figure 3C). In addition, the application of calpeptin or PI also significantly inhibited the effect of NS1643 on Cav-1 dephosphorylation and PTP1B cleavage (Figure 3D). These data indicate that the pharmacological activation of Kv11.1 potassium channel stimulates Ca^2+^/Calpain/PTP1B-dependent dephosphorylation of Cav-1, as shown in Figure 3E.

We next assessed grade 3 lymph node-positive breast cancer patient survival data. As shown in Figure 3F,G, respectively, we observed a clear pattern of high Cav-1 expression level that correlated with low survival, whereas high PTP1B correlated with better survival. This observation would be consistent with the above biochemical evidence, and together, may indicate PTP1B-mediated dephosphorylation of Cav-1, and thereby, reduced cancer cell migration and metastasis can increase survival in women with metastatic breast cancer.

### 3.4. Kv11.1 Channel Activation Promotes the Accumulation of β-catenin at Cellular Junctions that Are Dependent on Cav-1

Next, we assessed the underlying mechanism of decreased cell motility following Kv11.1 activation. We previously demonstrated that NS1643 induces β-catenin colocalization with its natural binding partner in epithelial cell adherens junctions, E-cadherin. Although MDA-MB-231 triple-negative breast cancer cells lack E-cadherin expression, we still observed an increase in active β-catenin (Figure 4A) and its accumulation (Figure 4B) at cell–cell junctions. Interestingly, this was abolished in Cav-1 depleted cells (Figure 4C).

The accumulation of β-catenin at cell junctions is a dynamic process, wherein β-catenin interacts with the actin cytoskeleton and travels back and forth to the surface membrane during the formation, maturation, disruption, and restoration of adherens junctions [36,37,38]. Fluorescent β-catenin-Venus cDNA was transduced in MDA-MB-231 cells, which were then stimulated with NS1643 for the live cell tracking of β-catenin using confocal microscopy. We observed that upon the addition of NS1643, the trafficking of intracellular β-catenin-positive vesicles and the displacement of these vesicles was significantly greater than in DMSO treated control cells (Figure 4D) suggesting that the activation of Kv11.1 may upregulate the movement of cytosolic β-catenin and thereby promote its junctional accumulation.

**Figure 4 cells-11-02461-f004:**
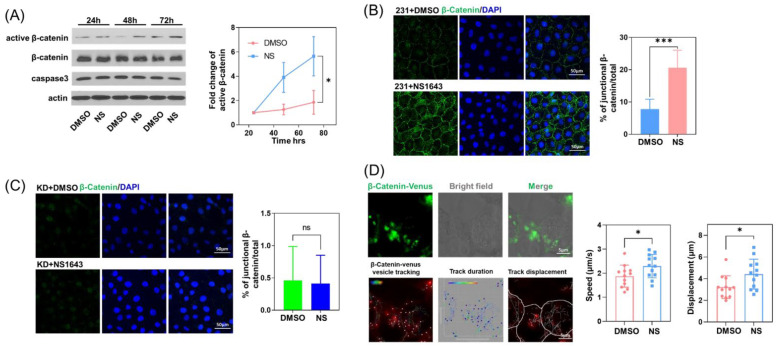
β-catenin at cell junctions increased in the presence of Kv11.1 activator. (**A**) MDA-MB-231 cells treated with DMSO or 50 μM NS1643 for 24, 48, or 72 h were lysed and subjected to Western blotting. NS treatment did not change apoptosis marker caspase-3, but increased the level of active β-catenin, n = 3. **p* < 0.05 via ANOVA. (**B**) MDA-MB-231 cells treated with DMSO or 50 μM NS1643 for 72 h were fixed and immunostained for β-catenin. Cells were visualized using confocal microscopy. Images were processed using ImageJ. Junctional β-catenin staining increased in NS1643 treated cells, n = 10. (**C**) Cav-1 depleted MDA-MB-231 cells treated with DMSO or 50 μM NS1643 for 72 h were fixed and immunostained for β-catenin. No change in junctional β-catenin staining was detected upon NS1643 treatment in the absence of Cav-1, n = 10. (**D**) MDA-MB-231 cells transfected with β-catenin-Venus were treated with DMSO or 50 μM NS1643 and visualized via confocal microscopy for 30 s. The movement of fluorescent β-catenin vesicles was tracked using Imaris. β-catenin-positive vesicle velocity was increased by NS1643. Total displacement representing the translocation of β-catenin-positive vesicles was also increased by NS1643, n = 10. All data are mean ± SD; ns = not significant * *p* < 0.05; *** *p* < 0.001 via *t* test.

### 3.5. Cav-1 Binds to β-Catenin in a Phosphorylation-Dependent Manner and Prevents Its Junctional Accumulation

After demonstrating that the junctional accumulation of β-catenin induced by Kv11.1 is Cav-1 dependent, we next assessed whether this is regulated by Cav-1 phosphorylation. First, we evaluated the association between Cav-1 and β-catenin by immunoprecipitation. In cells treated with NS1643 for 1 h, co-IP of Cav-1 and β-catenin decreased slightly, but after 24 h, it dropped by over 80%, as shown in Figure 5A. We then introduced WT, phosphomimicking Y14D, or phosphodefective Y14F-Cav1 mutants into MCF-7 cells (which have low endogenous Cav-1 expression). After treatment with NS1643, we only detected the disassociation of Cav-1 and β-catenin in WT-Cav-1 expressing cells. Consistent with the interaction being dependent on Cav-1 phosphorylation, we observed a low basal interaction of Y14F-Cav1 and β-catenin, and no effect of NS1643. Further, cells expressing phosphomimicking the Y14D Cav-1 mutant showed greater co-IP of the two proteins, and were also not affected by KV11.1, as shown in Figure 5B. Therefore, the interaction between Cav-1 and β-catenin is phosphorylation dependent.

We also co-transfected CFP-tagged Cav-1 mutants and β-catenin-Venus into Cav-1 depleted MDA-MB-231 cells. The colocalization of fluorescent Cav-1 and β-catenin was higher in Y14D-Cav-1 expressing cells, as compared to WT and Y14F-Cav-1 expressing cells (Figure 5C). Further, only the WT-Cav-1 interaction with β-catenin was diminished by NS1643 (Figure 5D) which is consistent with Western blot results. Therefore, the pharmacologic activation of Kv11.1 induces the dephosphorylation of Cav-1, which promotes its disassociation from β-catenin, enabling β-catenin to accumulate at cell–cell junctions. This result strongly suggests that the junctional accumulation of β-catenin contributes to adhesions, which may reduce cell motility and inhibit cancer cell migration, even in the absence of E-cadherin. To test this hypothesis, we tracked single cell migration in MCF7 cells transfectedd with the WT or phospho-mutant forms of Cav-1. Consistent with the hypothesis, reduction in cell motility caused by NS1643 treatment was observed in WT Cav-1 expressing cells, whereas Y14F-Cav1 reduced cell migration and Y14D-Cav1 increased cell migration independent of NS1643 treatment (Figure 5E–G and Appendix A).

### 3.6. Kv11.1 Activation Enhances β-catenin Interaction with Adhesion Complexes to Promote contact Inhibition

It was previously demonstrated that E-cadherin/β-catenin interaction occurs in the endoplasmic reticulum (ER) [39], and the disassociation of E-cadherin/β-catenin leads to the rapid degradation of β-catenin in the proteosome [40,41]. The junctional binding partners for β-catenin in triple-negative MDA-MB-231, which lack E-cadherin expression, are not clear. To reveal the potential β-catenin binding partners, we performed a mass spectrometry analysis of cellular proteins that co-immunoprecipitate with β-catenin antibody from MDA-MB-231 cells treated with vehicle (DMSO) or NS1643. Ingenuity Pathway Analysis revealed a series of signaling proteins, to which the interaction with β-catenin was upregulated by NS1643 treatment. Of note, pathways associated with cell morphology and migration (actin cytoskeleton, integrins, Rho family GTPases and its regulators, and signaling associated with endocytosis; Figure 6A) were increased, which is consistent with our observations showing an elevated level of β-catenin accumulation at the membrane. The summarized machine learning pathway analysis suggests that these changes primarily affect cell migration. We also found enhanced β-catenin association with a variety of proteins affiliated with the cytoskeleton (Figure 6B), which again is consistent with the significant impact observed on cell motility. Importantly, we successfully identified a potential binding partner for β-catenin in the cadherin family, namely R-cadherin. Although whole cell protein expression level of R-cadherin was not affected by NS1643, its binding to β-catenin was significantly elevated by NS1643 (Figure 6C,D). Remarkably, NS1643 also provoked an increased interaction of β-catenin with several desmosomal proteins, including plakophilin, desmoplakin, and desmoglein-2 (Figure 6C,E). These data suggest that the activation of the Kv11.1 potassium channel promotes β-catenin accumulation at cell–cell junctions in association with multiple adhesion complexes, thereby reducing cell motility. 

## 4. Discussion

Kv11.1, a voltage-gated potassium channel expressed in the heart, neurons, smooth muscle cells, and endocrine cells, was previously observed to be abundant in a large variety of primary and metastatic tumor cells [42]. Interestingly, it was not found in healthy tissue from patients from which the tumors were derived [10], which makes it a potential target for cancer therapy. Interestingly, while the pharmacologic inhibition of Kv11.1 is associated with drug-induced cardiac arrhythmia due to long QT syndrome [43], the activation of Kv11.1 can cause shortening of the cardiac action potential, although no severe adverse effects were observed [44], highlighting the possibility for utilizing Kv11.1 activators as anti-cancer drugs. In our previous work, we have shown that the activation of Kv11.1 inhibits tumor growth and metastasis both in vitro and in vivo [45] without serious side effects; the underlying mechanism, however, was unclear.

Although its role in cancer has been a long-standing controversial topic, caveolin-1 (Cav-1) has established its importance in almost every cancer-related theme. Our study revealed that the inhibitory role of Kv11.1 activation on cell migration is dependent on Cav-1 expression, and more specifically, on its dephosphorylation. When Cav-1 is downregulated, the inhibition of both single cell and collective cell migration is no longer observed. However, we did detect elevated cancer cell growth in Cav-1 depleted breast cancer cells, similar to a previous report [46]. Both wound closure rate and spheroid volume (data not shown) were higher in Cav-1 depleted cells when compared to naïve cells. Perhaps more important is the correlation of Cav-1 tyrosine phosphorylation with enhanced tumor cell migration and metastasis [23]. We observed that Cav-1 Tyr14 phosphorylation is rapidly and progressively reduced by the treatment of cells with the Kv11.1 activator NS1643. We therefore hypothesized that the dephosphorylation of Cav-1 induced by the pharmacologic Kv11.1 activator NS1643 may be a novel mechanism for regulating cancer cell migration and metastasis.

After excluding the involvement of Src kinase inhibition, we focused on tyrosine phosphatases and the mechanism of Cav-1 dephosphorylation induced by Kv11.1 activation. Based on our previous findings, which showed that NS1643 promotes Ca^2+^ influx [35], we assessed the correlation between Ca^2+^ influx and Cav-1 dephosphorylation. Our results suggest that Kv11.1 activation-dependent Cav-1 dephosphorylation is triggered by K^+^ efflux/Ca^2+^ influx and the activation of the protease calpain. Activated calpain cleaves and activates PTP1B, which can then dephosphorylate Cav-1 tyrosine 14, as shown in Figure 3E. PTP1B was previously shown to bind and dephosphorylate Cav-1 and thereby decrease phospho-Cav-1-mediated breast cancer cell migration [16,47]. Thus, PTP1B and Cav-1 expression appear to have apposing roles in mediating breast cancer metastasis. Consistent with our hypothesis and with the survival of patients with grade 3 lymph node-positive breast cancer, greater Cav-1 gene expression is associated with poor survival while increased PTP1B gene expression correlates with a better overall survival (Figure 3F,G).

Next, we explored the mechanism by which the NS1643-induced dephosphorylation of Cav-1 inhibits cancer cell migration. Although the phosphorylation of Cav-1 was shown to promote cell migration by facilitating actin remodeling, focal adhesion turnover, pseudopodial protrusions, and stress fiber formation [25,48], in a previous study, we showed that the activation of Kv11.1 significantly increases β-catenin levels at the junctional membrane in MCF-7 cells [9]. Here, we observed the same effect of NS1643 in triple negative MDA-MB-231 cells, which was absent in Cav-1 depleted cells. Membrane targeted β-catenin associates with adhesion complexes at sites of cell–cell contact. There are three types of adhesive junctions in epithelia, tight junctions (TJ), adherens junctions (AJ), and desmosomes [49]. Catenins are often found in AJs, which are highly dynamic and cycle through initiation, spreading, maturation, and turnover programs [38]. Catenins bind to cadherins, which exhibit continuous recycling in vesicles between the plasma membrane and intracellular compartments. Enhanced β-catenin localization in the membrane at sites of cell–cell contact was observed in NS1643 treated cells, suggesting enhanced formation and dynamics of adherens junctions.

Cadherins and catenins form complexes during the early stages of their biosynthetic process, namely in the ER, which then localize to cell–cell contacts to maintain polarized cellular orientation, the integrity of barriers, and contact inhibition [50]. The disassociation of cadherin/catenin complexes can lead to the degradation of catenins [40,41]. The triple negative breast cancer cell line MDA-MB-231 has undergone an epithelial-to-mesenchymal transition (EMT), and lacks E-cadherin, P-cadherin, and N-cadherin expression [51,52,53]; and thus, the adhesive junctional membrane binding partner of β-catenin in MDA-MB-231 cells was heretofore unclear. Taking advantage of tandem mass tag (TMT)-based quantitative proteomics, we identified a possible binding partner of β-catenin: R-cadherin (Cadherin-4), which was shown to be a tumor suppressor in breast, colorectal, and gastric cancer [54,55,56]. Although we did not detect a change in total R-cadherin protein expression in NS1643 treated cells, we did observe an increase in the interaction between R-cadherin and β-catenin. In vascular smooth muscle cells, R-cadherin was found in a complex with β-catenin and was able to inhibit β-catenin nuclear translocation [57]. In MDA-MB-231 cells, R-cadherin downregulation diminished the junctional accumulation of β-catenin [55]. Interestingly, we observed that β-catenin binds to Cav-1 in a phosphorylation-dependent manner, and therefore, it seems that Cav-1 and R-cadherin are competitive binding partners for β-catenin. When Cav-1 is phosphorylated (i.e., in absence of NS1643 treatment), free β-catenin binding to Cav-1 is favored, such as during caveolae-mediated endocytosis and the recycling of AJs. Upon the EGF stimulation of normal and tumor-associate pancreatic epithelial cells, for example, AJs quickly undergo disassembly and endocytosis via caveolae [58] in a manner dependent on the phosphorylation of Cav-1 tyrosine 14 [31,59]. This may suggest that the Kv11.1 activator, by signaling Cav-1 tyrosine 14 dephosphorylation, promotes junctional β-catenin accumulation by inhibiting both the disassembly of AJs and the internalization of caveolae. Furthermore, blocking the disassembly of AJs enhanced cell–cell contacts also leads to increased β-catenin interactions with several other important cytoskeletal and junction-associated proteins. The junctional accumulation of β-catenin also prevents its translocation into the nucleus where it activates canonical Wnt signaling, which facilitates gene expression associated with cell proliferation, survival, and migration [60].

Among the proteins that exhibited increased association with β-catenin are several focal adhesion (FA) structural proteins, such as paxillin, vinculin, and focal adhesion kinase (FAK). By connecting the extracellular matrix to the cytoskeleton via integrins, FAs enable cell migration [61]. Meng and coworkers previously demonstrated the critical role of phospho-Cav-1 as a molecular switch that regulates FAK phosphorylation and focal adhesion dynamics, and thereby cell migration [62]. Whether β-catenin is involved in the phospho-Cav-1-dependent regulation of focal adhesion dynamics is unknown. Although there is no direct evidence showing that β-catenin localized to FAs per se, the pull-down of FA-associated proteins with β-catenin, as observed in our proteomics analysis, may indicate a crosstalk between the FA and AJ multiprotein complexes. The kinetics of Cav-1 phosphorylation/dephosphorylation versus alterations in AJ protein complexes and how this affects focal adhesion dynamics and cell migration may therefore be a very intriguing area for future study.

In addition to FA- and AJ-associated proteins, we detected an increase in the association of β-catenin with the desmosomal proteins desmoglein-2, desmoplakin, and plakophilin. Unlike R-cadherin, whole cell expression levels of these proteins increased, which theoretically could lead to an increase in desmosome and hemi-desmosome structures, which we did not assess. Hemi-desmosomes facilitate adhesion of the cell body to the matrix [63]. Increasing the number of hemi-desmosome structures could explain reduced single cell migration in the presence of NS1643, as this would be independent of cell–cell contact. Cav-1 was reported to be involved in the assembly and disassembly of desmosomes [64,65], but its association with desmosomal protein expression requires further investigation. 

## 5. Conclusions

In summary, in the present study, we showed that the activation of Kv11.1 promotes the dephosphorylation of Cav-1 and the release of β-catenin from Cav-1; and the increased association of β-catenin with R-cadherin, desmosome adhesion proteins desmoglein-2, desmoplakin, and plakophilin; as well as the FA proteins paxillin, vinculin, and FAK, resulting in reduced cell migration. The results also suggest that this novel underlying mechanism can account for previous observations in which a reduction in migration and metastasis in vitro and in vivo were observed after treatment with Kv11.1 activators [45]. The activation of Kv11.1 is not enough to kill cancer cells in situ. Furthermore, a recent study by our group showed that the dephosphorylation of Cav-1 promotes mitophagy and therefore may increase cancer cell survival by eliminating damaged mitochondria [30]. Taken together, these studies indicate that the Kv11.1 activation-dependent dephosphorylation of Cav-1 arrests cancer cell migration, and thereby substantiates its potential clinical use for the prevention of metastasis. To maximize its cancer-killing effect, Kv11.1 activators could be combined with mitophagy or autophagy blockers to reduce Cav-1 dephosphorylation-dependent survival signaling. 

## Figures and Tables

**Figure 3 cells-11-02461-f003:**
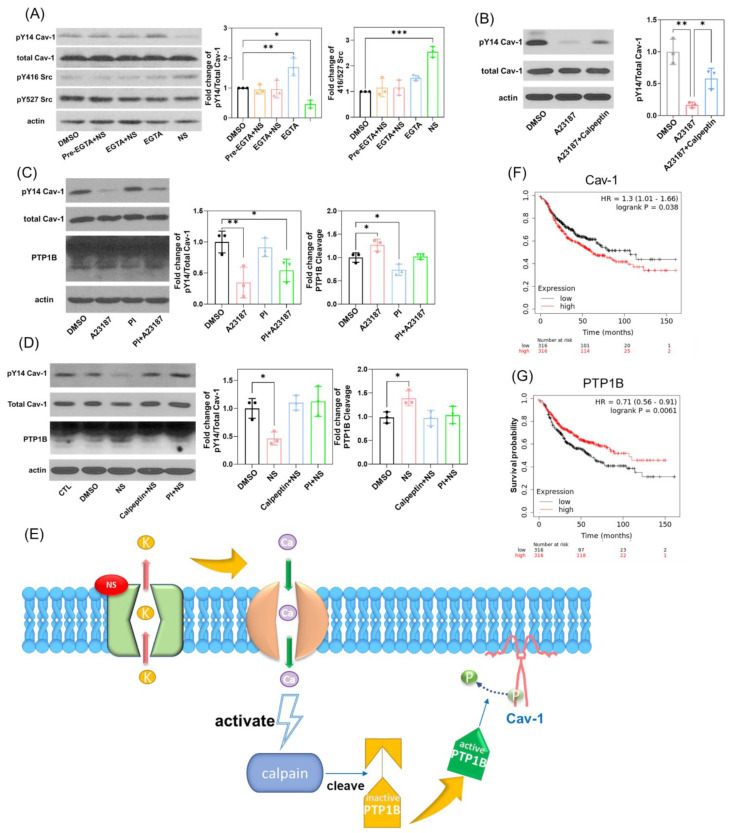
Kv11.1 activator NS1643 activates tyrosine phosphatase PTP1B, resulting in dephosphorylation of Cav-1. (**A**) MDA-MB-231 cells treated with DMSO, 50 μM NS1643, and 10 μM EGTA for 30 min were lysed and subjected to Western blotting. Lane 2 are cells pretreated with 10 μM EGTA for 15 min before adding NS1643. EGTA treatment prevented Cav-1 dephosphorylation and Src activation induced by NS1643, suggesting that the effect is calcium dependent. (**B**) MDA-MB-231 cells treated with DMSO, 5 μM calcium Ionophore A23187, and 20 μM calpain inhibitor calpeptin for 30 min were lysed and subjected to Western blotting. Calpeptin partly blocked Cav-1 phosphorylation induced by A23187. (**C**) MDA-MB-231 cells treated with DMSO, 5 μM calcium Ionophore A23187, and 16 μM PTP1B inhibitor (PI) for 30 min were lysed and subjected to Western blotting. A23187-induced Cav-1 dephosphorylation was partly blocked by the inhibitor of PTP1B, which is a downstream tyrosine phosphatase activated by calpain-dependent cleavage. (**D**) MDA-MB-231 cells treated with DMSO or 50 μM NS1643 alone or in combination with 40 μM calpeptin or 16 μM PTP1B inhibitor for 30 min were collected, lysed, and subjected to Western blotting. Both calpeptin and PTP1B inhibitor restored the phosphorylation of Cav-1 and prevented PTP1B cleavage induced by NS1643, n = 3–4. All data are mean ± SD; * *p* < 0.05, ** *p* < 0.005, *** *p* < 0.001 via ANOVA. (**E**) Signaling of Kv11.1 induced dephosphorylation of Cav-1. Kaplan–Meier survival analysis in grade 3 lymph node-positive breast cancer patients relative to Cav-1 (**F**) and PTP1B (**G**) mRNA expression.

**Figure 5 cells-11-02461-f005:**
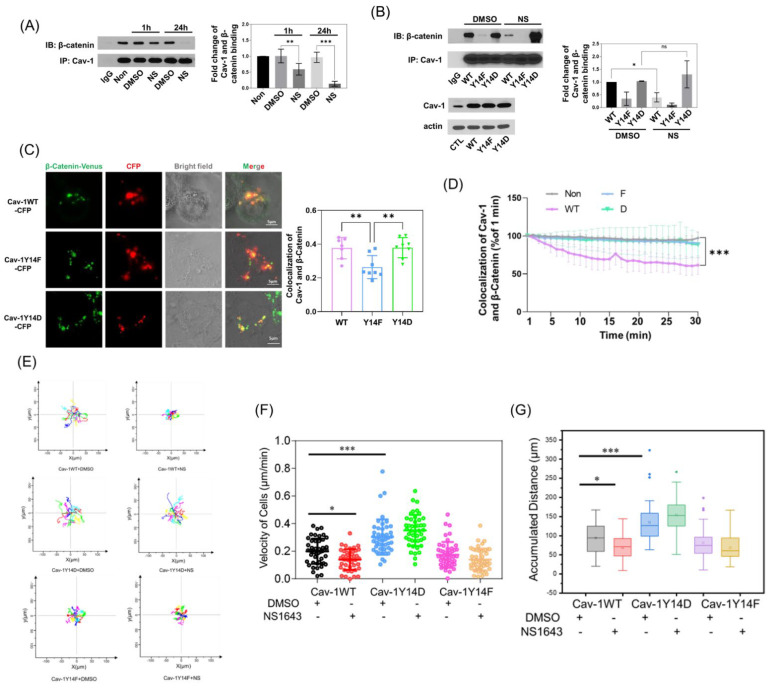
β-catenin interacts with Cav-1 in a phosphorylation-dependent manner. (**A**) MDA-MB-231 cells treated with DMSO or 50 μM NS1643 for 1 or 24 h were lysed and immunoprecipitated with anti-Cav-1 antibody. The pellets were washed, eluted, and subjected to Western blotting. Results suggest that NS1643 inhibited the interaction between Cav-1 and β-catenin, n = 3. (**B**) MCF-7 cells were transfected with Cav-1 wild type (WT), phosphodefective Y14F, and phosphomimicking Y14D mutants; and treated with DMSO or 50 μM NS1643 for 1 h. Cells were then lysed and immunoprecipitated with anti-Cav-1 antibody. The beads were then washed and boiled. The eluted proteins were blotted for β-catenin. Co-IP of Cav-1 and β-catenin was inhibited by NS1643 in Cav-1 WT expressing cells, but not with Y14F mutant, and remained high in Y14D mutant expressing cells, indicating that their interaction is phosphorylation-dependent, n = 3. (**C**) Cav depleted MDA-MB-231 cells were co-transfected with CFP-tagged WT, Y14F, and Y14D Cav-1 mutants and β-catenin-Venus for 48 h. Live cells were then visualized using confocal microscopy. The colocalization of CFP-tagged Cav-1 mutants and β-catenin-Venus was reduced in Cav-1Y14F mutant expressing cells, n = 8. (**D**) The colocalization of CFP-tagged Cav-1 mutants and β-catenin-Venus was assessed using time lapse confocal image sequences for 30 min. The colocalization of Cav-1 and β-catenin was reduced by NS1643 only in Cav-1 WT expressing cells, n = 6. (**E**) MCF-7 cells transfected with CFP-tagged mutants were treated with DMSO or 50 μM NS1643 and imaged using fluorescence microscopy every 6 min to track the movement of individual cells for 4 h. Image sequences were processed using ImageJ. Representative cell tracks are displayed. (**F**) The motility of cells expressing the CFP signal was measured. The inhibitory effect of NS1643 on cell migration was only detected in Cav-1WT expressing cells, n ≥ 35. (**G**) The accumulated distance moved by cells expressing CFP was only affected in Cav-1WT expressing cells, n ≥ 35. All data are mean ± SD; * *p* < 0.05, ** *p* < 0.005, *** *p* < 0.001 via ANOVA.

**Figure 6 cells-11-02461-f006:**
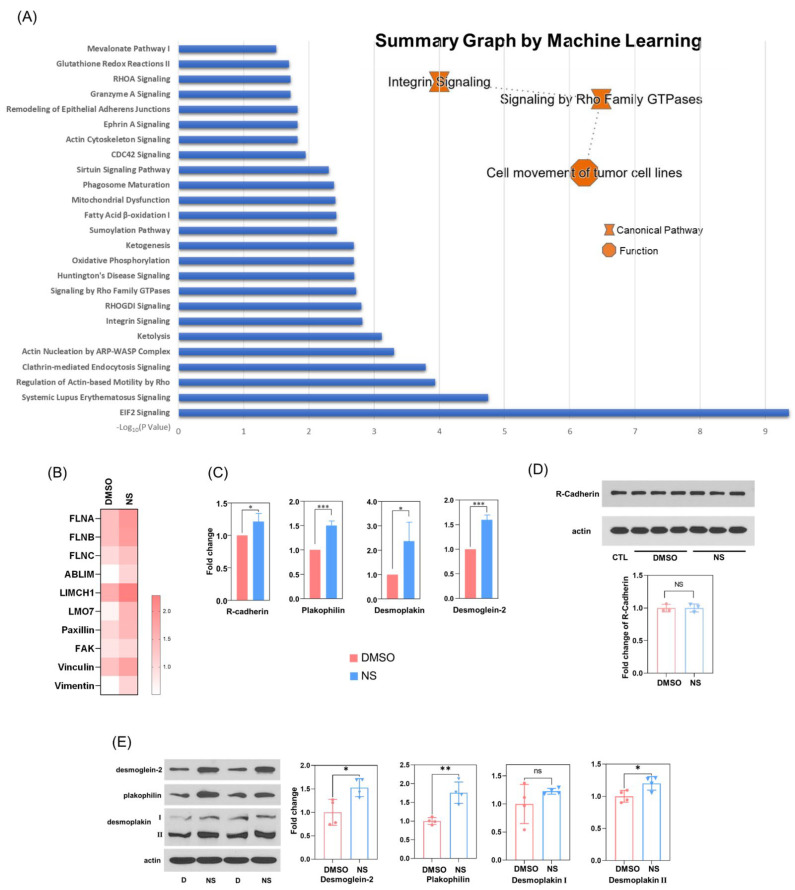
Kv11.1 activation promotes β-catenin interaction with R-cadherin at sites of cell–cell contact. (**A**) MDA-MB-231 cells treated with DMSO or 50 μM NS1643 for 48 h were lysed and immunoprecipitated with an anti-β-catenin antibody covalently conjugated with magnetic beads. The pellets were washed, eluted, and labeled with TMT6plex Isobaric Label Reagent Set (as Appendix A). Labeled samples were analyzed via mass spectrometry. Protein intensity was normalized to β-catenin intensity. Data were processed using Ingenuity Pathway Analysis (IPA) and a summary graph was generated using machine learning prediction. (**B**) Heat map of cytoskeleton-associated proteins calculated from the results of TMT labeling proteomics. Association of 11 proteins with β-catenin, which include actin crosslinking proteins and those associated with focal adhesion complexes increased in cells treated with NS1643. (**C**) Fold change of junctional proteins interacting with β-catenin in DMSO, and NS1643 treated cells from proteomics analysis. R-cadherin and desmosome-related proteins were shown to interact with β-catenin, and pulldown of these proteins increased in cells treated with NS1643, n = 3. (**D**) MDA-MB-231 cells that were untreated (CTL) or treated with DMSO or 50 μM NS1643 for 24 h were lysed and subjected to Western blotting. No change in R-cadherin expression was found in NS1643 treated cells, n = 3. (**E**) The expression of desmoglein-2, plakophilin, and desmoplakin-2 increased in cells treated with NS1643, whereas no effect was observed for desmoplakin-1 expression, n = 4. All data are mean ± SD, ns = not significant; * *p* < 0.05, ** *p* < 0.005, *** *p* < 0.001 via *t* test.

## Data Availability

Proteomics data have been deposited in Zenodo, which can be accessed by the link: https://doi.org/10.5281/zenodo.6761903, accessed on 5 August 2022. All other data needed to evaluate the conclusions in the paper are present in the paper or in the Appendix A. Data and materials generated for this study are available from the corresponding author upon reasonable request.

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
