# Peer review of "Pharmacological Activation of Potassium Channel Kv11.1 with NS1643 Attenuates Triple Negative Breast Cancer Cell Migration by Promoting the Dephosphorylation of Caveolin-1"

_cells, 2022, doi:10.3390/cells11152461_

Round 1

Reviewer 1 Report

The field is interesting and the results consistent with conclusions but the manuscript should be ameliorated. My specific concerns start from the title. Given that NS1643 is the only Kv11.1 activator molecule used, it should be mentioned in the title as well as it should be specified that the work focused on TNBC specifically. In this regard, it remains to determine if the observations are limited to the selected cells line or breast cancer tumor type. In addition, what about the other Kv11.1 activator molecule PD115087 also mentioned as able to inhibit breast cancer metastasis? Did the authors verified its ability to act similarly to NS1643? It should offer a further pharmacological opportunity considering that non obvious translation from in vitro to in vivo studies. The Introduction section should synthesized and rendered more focused on the background concepts needed for understand the aim and the study design. Regarding the Results section, some data should be moved in Supplementary files to simplify the reader understanding (i.e. Fig 2D-E-F). The occasional use of ER+ BC cells MCF7 needs to better explained. All figure legend need to be edited by avoid to repeat the same description for different graphs (in the same figure).

Reviewer 2 Report

The paper covers an impressive research which relates the action of Kv11.1 channels with the dephosphorylation of the Cav-1, which in turn leads to reduced motility of cancer cells and reduced metastatic potential. Some minor remarks:

- the content of Figs. 3E, 6A is unreadable in PDF format - resolution is too small.
- the data is usually presented as mean+/-SD - in cases of overlapping uncertainty it would be interesting to see also SD of the mean.
- on page 7, line 269 citation is missing.
- on page 8 - the Kaplan Meier plots could be tested for statistical significance - or at least some error bars for probability estimates could be provided. Also the protective effect could be quantized in form of a number.
- on page 11, Fig. 6D it is mentioned that machine learning was used to obtain the results - what methods / software?
- on page 13, in the discussion one finds that the authors propose the main effect of increased catenin levels to be provided by the catenin-R-cadherin interaction. Is it possible to provide some hints on the possible percentage contribution by other signaling pathways? I.e. does cadherin downregulation by certain percentage cause a downregulation of catenin by the same amount or by some (large?) fraction of this percentage?
